# Murine modeling of menstruation identifies immune correlates of protection during *Chlamydia muridarum* challenge

Laurel A. Lawrence[1], Mark Elliott Williams[1], Paola Vidal[1], Richa S. Varughese[1], Zheng-Rong Tiger Li[1], Thien Duy Chen[1], Melissa A. Roy[2], Steven C. Tuske[1], Anice C. Lowen[1], Christopher D. Scharer[1], William M. Shafer[1,3], Alison Swaims-Kohlmeier[1,4,5]*

1 Department of Microbiology and Immunology, Emory University School of Medicine, Atlanta, Georgia, United States of America, 2 Division of Pathology Emory National Primate Research Center, Emory University, Atlanta, Georgia, United States of America, 3 Laboratories of Bacterial Pathogenesis, Atlanta Veterans Affairs Medical Center, Decatur, Georgia, United States of America, 4 Department of GYNOB, Emory University School of Medicine, Atlanta, Georgia, United States of America, 5 Division of HIV Prevention Centers for Disease Control and Prevention, Atlanta, Georgia, United States of America

* askohlm@emory.edu

## Abstract

The menstrual cycle influences the risk of acquiring sexually transmitted infections (STIs), including those caused by the pathogen *Chlamydia trachomatis* (*C. trachomatis*). However, the underlying immune contributions are poorly defined. A mouse model simulating the repetitive immune-mediated process of menstruation could provide valuable insights into tissue-specific determinants of protection against chlamydial infection within the cervicovaginal and uterine mucosae of the female reproductive tract (FRT). Here, we used the pseudopregnancy approach for inducing menstruation in naïve C57Bl/6 mice and performed vaginal challenge with *Chlamydia muridarum* (*C. muridarum*) over the course of decidualization, endometrial tissue remodeling, and menstruation. This strategy identified that a time point over pseudopregnancy corresponding to the late luteal phase of the menstrual cycle correlated with reduced bacterial burden. By evaluating the early infection site following challenge at this time point, we found that a greater abundance of NK cell populations and proinflammatory signaling, including IFNγ, were strongly correlated with protection. FRT immune profiling in uninfected mice over pseudopregnancy or in pig-tailed macaques over the menstrual cycle identified periodic NK cell infiltration into the cervicovaginal tissues and luminal surface occurring over a similar time frame. Notably, these cell populations were transcriptionally distinct and enriched for programs associated with NK cell effector functions. Depletion of FRT NK cells during the late luteal phase time frame resulted in a loss of protection, enabling productive infection following *C. muridarum* challenge. This study shows that the pseudopregnancy murine menstruation model recapitulates dynamic changes occurring in mucosal immune states throughout the

**Data availability statement:** Sequencing data has been submitted to the Gene Expression Omnibus (GEO) repository: GSE297609.

**Funding:** This study was supported by NIH grant R21AI180610 (A.S-K). (Cytokines) This study was supported in part by the Emory Multiplexed Immunoassay Core (EMIC), which is subsidized by the Emory University School of Medicine and is one of the Emory Integrated Core Facilities. Additional support was provided by the National Center for Georgia Clinical & Translational Science Alliance of the National Institutes of Health under Award Number UL1TR002378. (Pathology) This research is supported by the Emory National Primate Research Center National Primate Research Center Grant No. ORIP/OD P51OD011132 . (NK cell Sorting) Research reported in this publication was supported in part by the Pediatrics/Winship Flow Cytometry Core of Winship Cancer Institute of Emory University, Children's Healthcare of Atlanta and NIH/NCI under award number P30CA138292. The content is solely the responsibility of the authors and does not necessarily represent the official views of the National Institutes of Health. W.M.S. is a recipient of a Senior Research Career Scientist award from the Medical Research Service of the Department of Veterans Affairs. The content is solely the responsibility of the authors and does not necessarily reflect the official views of the National Institutes of Health. the Centers for Disease Control and Prevention, or the Department of Veterans Affairs. The funders had no role in study design, data collection and analysis, decision to publish, or preparation of the manuscript.

**Competing interests:** The authors have declared that no competing interests exist.

FRT as a result of endometrial remodeling and identifies NK cell localization at the FRT barrier site of pathogen exposure as essential for immune protection against primary *C. muridarum* infection.

## Author summary

Although the vast majority of women and adolescent girls of reproductive age experience menstruation, we have limited insight into how this tissue remodeling process alters mucosal immune defenses against infection by genitourinary pathogens. In this study, we used a murine model of menstruation to investigate how endometrial shedding and repair alters the FRT immune landscape to influence chlamydial infections. Using this approach, we identified that endometrial remodeling regulates a substantial pro-inflammatory immune response, including periodic NK cell recruitment into the cervicovaginal tissues. Transcriptional profiling showed that these cells were distinct from FRT NK cells in MPA-treated mice and exhibited an immune-activated state. The localization and enrichment of NK cells at the cervicovaginal barrier were determined to be responsible for providing rapid immune protection that reduced *C. muridarum* burden, as experimental depletion of these cells at this time point led to productive infections. Taken together, this study identifies that murine models of menstruation can be a valuable tool for investigating how the menstrual cycle modulates immune homeostasis and for identifying ways to strengthen mucosal immune defenses against genitourinary pathogens in women.

## Introduction

*Chlamydia Trachomatis* (*C. trachomatis*) infections spread through sexual contact and can result in severe diseases in women and congenitally infected newborns. *C. trachomatis* is the causative agent of one of the most common and costliest bacterial sexually transmitted infections (STIs) globally, with the majority of infections occurring in women and adolescent girls of reproductive age [1]. Although chlamydial infections remain an urgent global health issue, there is currently no vaccine that can protect against *C. trachomatis*. Notably, reinfections are common, which increases the likelihood of developing severe diseases, including pelvic inflammatory disease (PID), stillbirths, infertility, and an increased risk of acquiring more severe secondary infections such as those caused by *Neisseria gonorrhoeae* (*N. gonorrhoeae*) and human immunodeficiency virus type-1 (HIV) [2]. Although immune cells positioned within mucosal barrier sites, such as the cervicovaginal and uterine mucosae of the female reproductive tract (FRT), can provide an immediate effector response against invading pathogens due to their proximity at an infection site [3–5], these contributions to *C. trachomatis* infections of the FRT are unclear. Thus, greater insights into how the FRT tissue environments regulate cellular immune barrier defense against chlamydial infections are essential for informing prevention efforts.

Globally, the vast majority of women of reproductive age (15–49 years) experience periodic menstruation [6], and though dependent upon the immune system, the processes by which menstruation can influence protection against invading pathogens are poorly recognized. In regards to infection risk by the most prevalent bacterial STI pathogens in the U.S., including *C. trachomatis* and *N. gonorrhoeae,* it has been previously shown using animal modeling and human tissue samples that levels of the sex hormones progesterone and estrogen are associated with infection risk and the potency of immune effector responses [7–14]. However, we have limited insights into what role the menstrual cycle plays in determining these differences. A major challenge in studying how menstruation impacts immune defenses at mucosal barrier sites is a lack of model systems that menstruate [15]. In particular, access to common laboratory animal models in which immunologic and genetic approaches could facilitate mechanistic investigations into the dynamics of FRT tissue-localized immune cell populations would be ideally suited for this purpose [16–22].

Previously, a minimally invasive strategy for inducing menstruation in the BALB/c strain of inbred laboratory mice was reported using the pseudopregnancy method [23]. In this approach, BALB/c female mice in estrus were mated with vasectomized males to induce uterine decidualization. At the time frame of implantation, sesame seed oil is injected into the endometrial environment, leading to a state of terminal differentiation by decidual cells. The subsequent decline in progesterone causes rapid deterioration of the endometrium, which prompts uterine remodeling and discharge in the mice, similar to the process of menstruation occurring in species that naturally undergo spontaneous decidualization, such as humans and pig-tailed macaques [24,25].

To test whether the pseudopregnancy approach for inducing menstruation in mice might provide insights into tissue-specific immune determinates of *C. trachomatis* infection, we applied this method to the C57Bl/6 strain of inbred mice paired with vaginal challenge by *Chlamydia muridarum* (*C. muridarum)*, a murine strain of chlamydia which models both lower and upper FRT infection by *C. trachomatis* [26–28]. This strategy showed that following the induction of pseudopregnancy, C57Bl/6 mice exhibited progesterone fluctuations with corresponding innate immune cell recruitment into both the uterine and cervicovaginal mucosa followed by decidual discharge (*i.e.,* menses). We further discovered that pseudopregnant mice could undergo repeat cycling with corresponding progesterone and immune oscillations. By performing vaginal challenges with *C. muridarum* based on the time point of pseudopregnancy, we found that challenges administered during endometrial remodeling, specifically during the time frame progesterone withdrawal was detected in blood plasma, were unlikely to result in a productive infection. Immune profiling of the FRT tissues showed that endometrial remodeling was associated with increased IFNγ signaling and NK cell recruitment in the cervicovaginal tissues. To model whether this change occurs in naturally menstruating species, we used progesterone waveform reconstruction from female pig-tailed macaques of reproductive age and mice undergoing repeat pseudopregnancy and confirmed both increased IFNγ-associated signaling and NK cell infiltration at the cervicovaginal lumen during the luteal and late luteal phases. Transcriptomic profiling showed that FRT NK cells during progesterone withdrawal were enriched for inflammatory and cytotoxic programs consistent with an immune-activated state. Finally, to confirm the role of NK cells in early protection against primary *C. muridarum,* we depleted NK cells in mice during the time span of endometrial remodeling prior to vaginal challenge, which then resulted in productive chlamydial infections.

Taken together these data show that the menstrual cycle determines NK cell localization and transcriptional programming within the cervicovaginal mucosa, which plays an essential role in early immune protection against primary *C. muridarum* infection. Importantly, we demonstrate that the murine pseudopregnancy method for inducing menstruation is a valuable tool for investigating mucosal immune correlates of protection and risk against chlamydial infection and potentially for developing strategies that can strengthen mucosal immunity against genitourinary pathogens.

## Results

To investigate immune changes in the FRT occurring as a result of menstruation, we began by optimizing the murine pseudopregnancy approach to induce overt menstruation in C57Bl/6 mice. Female mice aged 6–12 weeks were mated

with vasectomized males, and successful ejaculation was confirmed by the detection of a vaginal plug the following morning (Fig 1A). The time points of endometrial remodeling (days 6–8), menstruation (days 10–11), and ovulation (days 12–14) over pseudopregnancy were determined by sex hormone measurements, vaginal cytology, uterine vascularization, and visible menstruation. First, by monitoring sex hormone levels from blood plasma over pseudopregnancy (Fig 1B), we observed a sharp progesterone increase on day 4 that peaked at day 6. Progesterone withdrawal due to the absence of fertilization was detectable on day 8, and by day 10, progesterone had decreased to a range consistent with day 2 of pseudopregnancy and similar to mice treated with the hormonal contraceptive medroxyprogesterone acetate (MPA) to control for reproductive cycling. In contrast to progesterone, estrogen levels generally remained within the range of those detected in MPA-treated mice, and mean levels did not significantly change until day 14, at which point estrogen sharply increased (Fig 1C). Next, to identify how pseudopregnancy impacted vaginal cellularity, we monitored changes by cytology (Fig 1D). Microscopy of vaginal smears over pseudopregnancy showed a large increase in leukocytes at days 6–8 (suggestive of increased inflammation), the detection of red blood cells between days 10–11, followed by a predominant population of anucleated vaginal epithelial cells consistent with ovulation on days 12–14 [29]. ALPHA-dri bedding, vaginal swabs, or visual inspection (Fig 1E) confirmed that mice were menstruating between days 10–11. Next, to measure uterine vascularization over pseudopregnancy, we performed intravital vascular (IV) labeling of circulating leukocytes prior to necropsy (Fig 1F and 1G) [16] and compared the frequency of circulating cells in the uterine horns of pseudopregnant mice with MPA-treated or sesame seed oil control mice (mice receiving an intrauterine injection with sesame seed oil but not mated with vasectomized male mice) (Fig 1H). This approach showed that starting on day 6, the mean levels of vascular (IV+) leukocytes began to increase and peaked on day 8, with 40% of the overall population, on average, representing cells from circulation. By day 12, the frequency of circulating cells was once again decreased into the range detected on day 2 of pseudopregnancy, similar to MPA and sesame seed oil control mice.

Although murine menstruation models, including the pseudopregnancy approach, have been previously described [23,30–32], whether laboratory mice could undergo repetitive menstruation similarly to humans has not been reported to our knowledge. Therefore, we tested whether the C57Bl/6 mice could undergo repeat menstruation by remating with vasectomized males immediately following menses for a period of 26 weeks and, in parallel, monitored daily progesterone concentrations (Fig 1I). This strategy showed that mice successfully underwent consecutive pseudopregnancy cycles as demonstrated by the detection of at least three successful matings immediately following menses and, importantly, the detection of corresponding progesterone oscillations that were consistent over time. To determine whether repeat cycles were functionally analogous, we compared uterine vascularization at day 8 of the first and second cycles (Fig 1J), which showed similar IV+ measurements. Overall, these data show that 1). the pseudopregnancy approach is reciprocated in C57Bl/6 mice and identifies key time points of endocrine-regulated endometrial remodeling and menstruation and 2). this approach models oscillatory uterine remodeling that occurs under menstrual cycle regulation.

Next, to investigate immune alterations at the FRT barrier as a result of endometrial remodeling, we performed immunofluorescent microscopy (Figs 2 and S1). By comparing tissue-localized myeloid cell (Gr-1+) populations, blood vessels (CD31+), and epithelial cells (EPCAM+) over pseudopregnancy, we observed pronounced changes occurring at both the uterine and cervicovaginal mucosae (Fig 2A-C). Consistent with our findings in Fig 1H, we identified increased vascularization of the uterine horns on day 8, identified by a robust increase in blood vessels. In regards to FRT localized cells, we found that following day 4 of pseudopregnancy, Gr-1+ immune cells were localized in the epithelial lining on day 6 and day 8. Notably, in the cervicovaginal (LFRT) tissues, immune cells were observed progressing through the epithelial barrier and aggregating at the apical surface within vaginal microfolds on day 6 and day 8 (Fig 2D). This behavior appeared distinct from the uterine tissues, where immune cells were more focused at the basal layer of the epithelium (decidua).

To better define cellular changes in the immune landscape over endometrial remodeling and menstruation in the FRT that might influence infection risk, we performed longitudinal profiling of tissue localized/IV negative (IV$_{neg}$) innate immune cell populations previously shown to be important for protection against primary chlamydial infections: neutrophils,

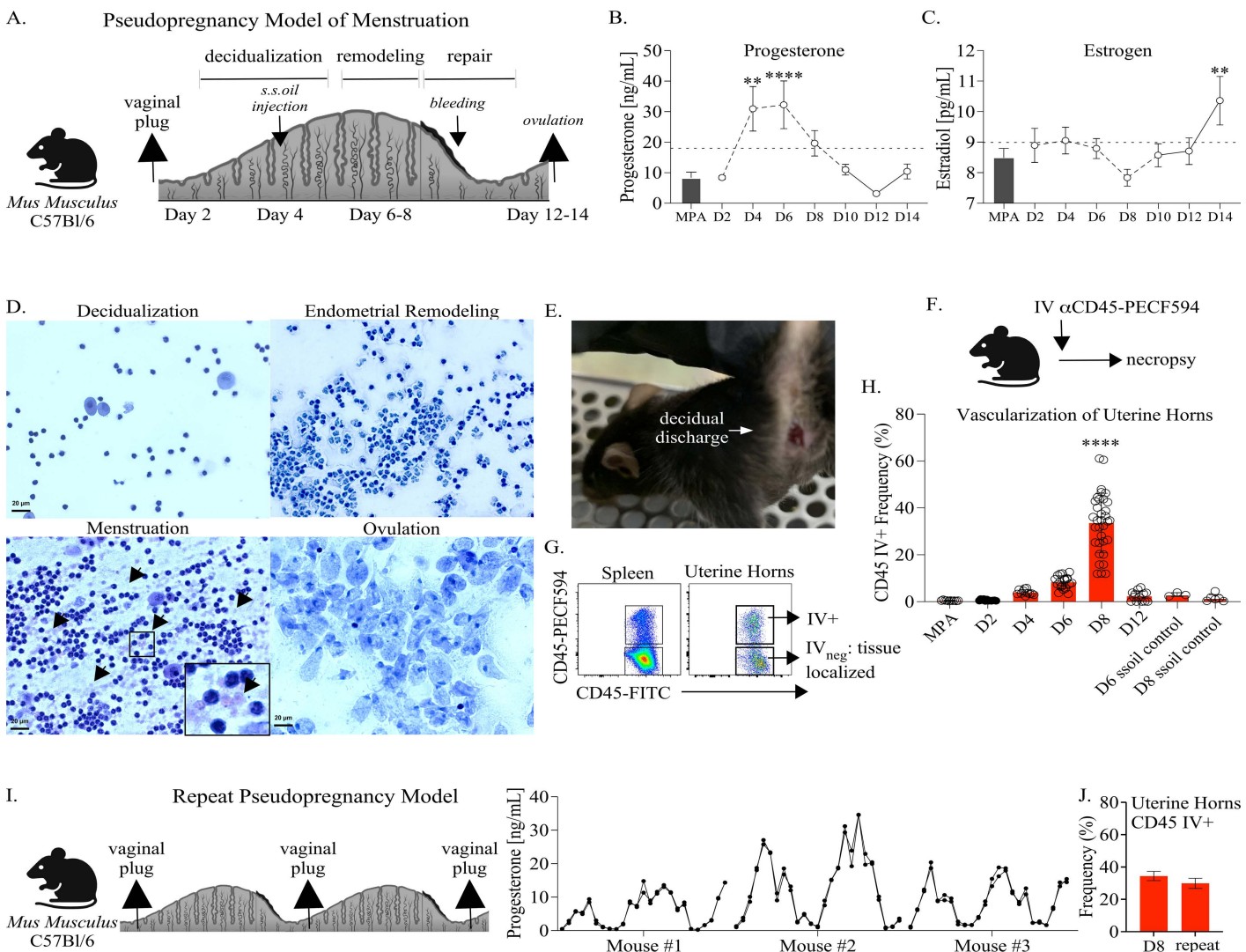

**Fig 1. Modeling cyclical menstruation in C57Bl/6 mice using the pseudopregnancy approach. (A).** Depiction of the pseudopregnancy approach for inducing menstruation in C57Bl/6 mice with the time frame of major endometrial changes emphasized. Sesame seed oil intrauterine injection is indicated as s.s.oil injection. **(B).** A mean symbol graph with the standard error of means (SEM) depicting progesterone or **(C).** estrogen levels measured from blood plasma over indicated time points of pseudopregnancy and plotted as concentration. Progesterone and estrogen levels from mice administered Medroxyprogesterone acetate (MPA) are shown as a bar graph with SEM for each comparison. **(B, C).** Models used to evaluate a mean deviation (horizontal dotted lines) were fit using one-sample t-tests. A minimum of 6 mice were measured at each time point. **(D).** Vaginal cytology over pseudopregnancy at indicated time points. Vaginal smears were stained using Hematoxylin and Eosin (H&E) and then visualized using microscopy at 40x magnification. Black arrows and side inset window (increased for visualization) identify red blood cells during menstruation. **(E).** A photo of a menstruating C57Bl/6 mouse following the induction of pseudopregnancy. Photo taken by A.S-K. **(F).** A schematic depicting the intravital vascular (IV) labeling approach for distinguishing leukocytes. **(G).** Cell flow plots from the spleen and uterine horns of a representative animal at day 8 of pseudopregnancy, illustrating the approach for distinguishing tissue-resident (IV$_{neg}$) or circulating leukocyte measurements (IV+) following IV labeling. Viable, singlet, leukocytes are distinguished by IV-labeling. **(H).** The frequency of IV+ leukocytes from uterine horns over the indicated time points of pseudopregnancy and compared with sesame seed oil injection control mice or mice treated with MPA. Each open circle represents an individual animal. Models used to compare a difference of means were fit using multiple comparisons with false discovery rate (FDR) testing: all comparisons with day 8 (D8) resulted in a significant difference of $p < 0.0001$. **(B, C, H).** p-values $\leq 0.05$ are shown as *$p \leq 0.05$, **$p < 0.01$, ***$p < 0.001$, ****$p < 0.0001$. **(I).** (Left panel) Depiction of the repeat pseudopregnancy approach in C57Bl/6 mice. (Right panel) A symbol graph depicting progesterone levels measured from the blood plasma of 3 mice over the course of consecutive pseudopregnancy cycles and plotted as concentration. Measurements are plotted in duplicate. **(J).** Bar graphs with SEM depicting the frequency of IV+ leukocytes from uterine horns at day 8 of the first (shown in H) and second (n = 10) cycle. Models used to compare a difference of means were fit using unpaired t-tests and were not significantly changed. **(A, F, I).** Created using BioRender.com.

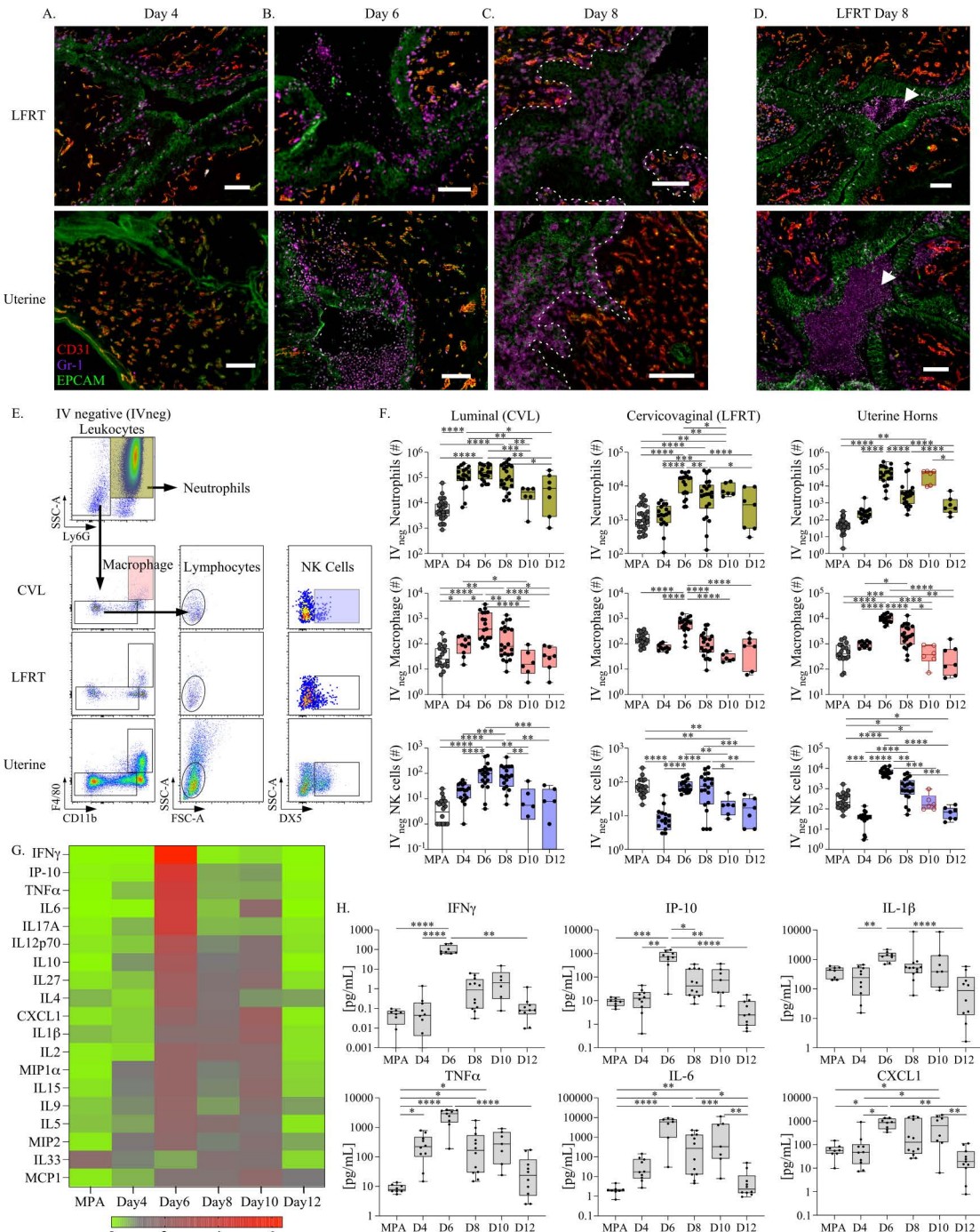

**Fig 2. Alterations in the lower and upper FRT immune landscape during endometrial remodeling. (A).** Fluorescence microscopy images taken at 20x magnification to visualize blood vessels (CD31 expression in red), myeloid cells (Gr-1 expression in purple), and epithelial cells (EPCAM expression in green) in the lower FRT (LFRT top panels), and uterine horns (bottom panels) at day 4, **(B).** day 6, and **(C).** day 8 of pseudopregnancy (dotted line indicates basal epithelium). **(D).** Additional images of the LFRT on day 8 show immune populations aggregating in the vaginal lumen (white arrows). (A-D). The white scale bars indicate 100 μm length **(E).** Flow cytometry cell gating strategy for measuring innate immune cells from anatomic compartments of the FRT. Viable singlet tissue-resident leukocytes are discriminated by the expression of Ly6G and side scatter characteristics from CVL, cervicovaginal tissue, and the uterine horns. The remaining populations are then measured for macrophage based on CD11b and F4/80 expression, and then NK cells are measured from F4/80 negative lymphocytes (based on size and granularity characteristics) followed by DX5 expression. **(F).** The total yield of tissue localized (labeled on the y-axis as negative for IV labeling: $IV_{neg}$) neutrophils (top panels), macrophage (center panels), and NK cells

(bottom panels) from FRT tissue sites are plotted as bar and whiskers graphs over pseudopregnancy and compared with mice administered MPA as a control. Day 10 uterine horn data points are emphasized in red to indicate potential menses contamination. **(G).** A heat map depicting the fold change in cytokines and chemokines measured from CVL over pseudopregnancy or MPA and ordered according to the greatest fold increase (top to bottom). **(H).** The concentrations of indicated cytokines are plotted as bar and whiskers graphs over pseudopregnancy and compared with mice administered MPA as a control. **(F, H).** Models used to compare a difference of means were fit using multiple comparisons with FDR testing: *p ≤ 0.05, **p < 0.01, ***p < 0.001, ****p < 0.0001.

macrophage, and NK cells (Fig 2E) [33–35]. To identify potential regional differences within and between FRT tissues, luminal cells were first collected by cervicovaginal lavage (CVL), followed by dissection of cervicovaginal (LFRT) tissues from the uterine horns. Single-cell suspensions were then assessed for immune cell populations using flow cytometry (Fig 2E). By quantifying tissue-resident leukocyte yields over pseudopregnancy (Figs 2F, S2 and S3), we identified that neutrophils increased within the uterine tissues at the peak of blood plasma progesterone levels (day 6 of pseudopregnancy), similar to previous reports [36,37]. While endometrial neutrophils have been previously implicated in facilitating endometrial shedding and repair, we also observed neutrophil infiltration into the LFRT occurring during this time frame. However, at the luminal surface (CVL), neutrophil numbers remained elevated from day 4 until menses onset and ovulation (days 10–12) and as compared to mice treated with MPA. We identified a similar trend with macrophage populations, which have been identified as important contributors to endometrial tissue breakdown and repair during menstruation [38]. These cells also sharply increased at day 6 in all of the FRT sites examined, including the luminal surface, and then contracted at days 10–12. Finally, we examined NK cells, which in the uterus are thought to play a critical role in implantation and have been shown to increase during the luteal phase when blood progesterone levels peak [39,40]. This analysis also showed cell increases occurring throughout the FRT on day 6 as compared with day 4; however, while neutrophil, macrophage, and NK cell numbers began to contract on day 8 in the uterine horns, elevated levels of luminal and LFRT NK cell numbers were sustained before ultimately decreasing on day 10. By comparing NK cells over pseudopregnancy with MPA treatment, we found that, while the number of NK cells in the LFRT tissues at days 6–8 were within a similar range, luminal NK cell numbers were significantly greater on days 6–8 of pseudopregnancy. To better understand the sustained elevation in luminal and LFRT NK cells at day 8 when leukocyte populations were contracting, we measured NK cell frequency within the total leukocyte pool (S4 Fig). This approach identified a significant enrichment in luminal NK cells at day 8 as compared with day 4 and day 6 of pseudopregnancy, and MPA treatment. This NK cell enrichment was also compartmentally restricted to the luminal surface, as NK cells in the underlying cervicovaginal tissues were instead most enriched in MPA-treated mice.

Next, we evaluated soluble immune mediators in cervicovaginal secretions over pseudopregnancy by measuring 19 pro-inflammatory cytokines and chemokines from CVL supernatants (Fig 2G and summarized in S1 Table). Corresponding to increases in cellular immune populations, most of these cytokines and chemokines exhibited elevations at day 6 of pseudopregnancy, with the greatest increases being IFNγ and the IFNγ-induced protein, IP-10. Because leukocyte infiltration during endometrial remodeling has been previously linked with increased uterine TNFα, IL1β, IL−8 (murine homolog: CXCL1), and IL-6 production [36,41], we specifically compared those CVL concentrations over pseudopregnancy in addition to IFNγ (Fig 2H). This analysis also showed sharp increases in the concentrations of these cytokines and chemokines at day 6 of pseudopregnancy, which by day 12 had decreased to levels detected within the ranges of day 4 and MPA. Although less well characterized in the context of the menstrual cycle, IFNγ signaling has been previously identified as important for facilitating implantation and pregnancy [42–44] and is predominantly produced by NK cells, suggesting similar regulation by the menstrual cycle [45]. Taken together, these data show that the LFRT and luminal surface also experience proinflammatory changes similar to the endometrium over pseudopregnancy. The only exception to these observations was the discovery of a sustained elevation of luminal NK cells during the time frame of progesterone withdrawal.

As the changes observed in NK cells and IFNγ signaling in the cervicovaginal environment over pseudopregnancy are, to our knowledge, less defined in menstruating species, we tested whether these discoveries were translationally relevant

by measuring these properties in pig-tailed macaques (**Fig 3A**). To perform a longitudinal analysis over the menstrual cycle, six female pig-tailed macaques of reproductive age were sampled bi-weekly to measure estrogen and progesterone from blood plasma and weekly for CVL collection for a period of 9 weeks. To stratify immune measurements, estrogen peaks (representing ovulation) and observed menstruation were used to determine cycle phases (**Fig 3B**). Using leukocyte-enriched CVL cells compared with PBMC (peripheral blood mononuclear cells), NK cells were assessed by flow cytometry (**Fig 3C**). To control for animal-to-animal variations, NK cell yields from each animal were calculated as a fold change over the course of the cycle. This approach showed that luminal NK cell numbers at the luteal phase (peak progesterone) and late luteal phase (detection of progesterone withdrawal) in macaques were increased, consistent with the sustained elevations observed during pseudopregnancy in mice. Following these peaks in NK cell yields, levels then decreased prior to menses onset. Next, we evaluated IFNγ signaling by measuring the IFNγ-induced protein, IP-10, which is generally found at higher concentrations and, thus, was more likely to fall within the range of assay detection for non-human primates (NHP) (**Fig 3D**). The levels of CVL IP-10 also peaked at the luteal phase, matching peak progesterone, followed by a rapid decrease, similar to the kinetics observed in mice.

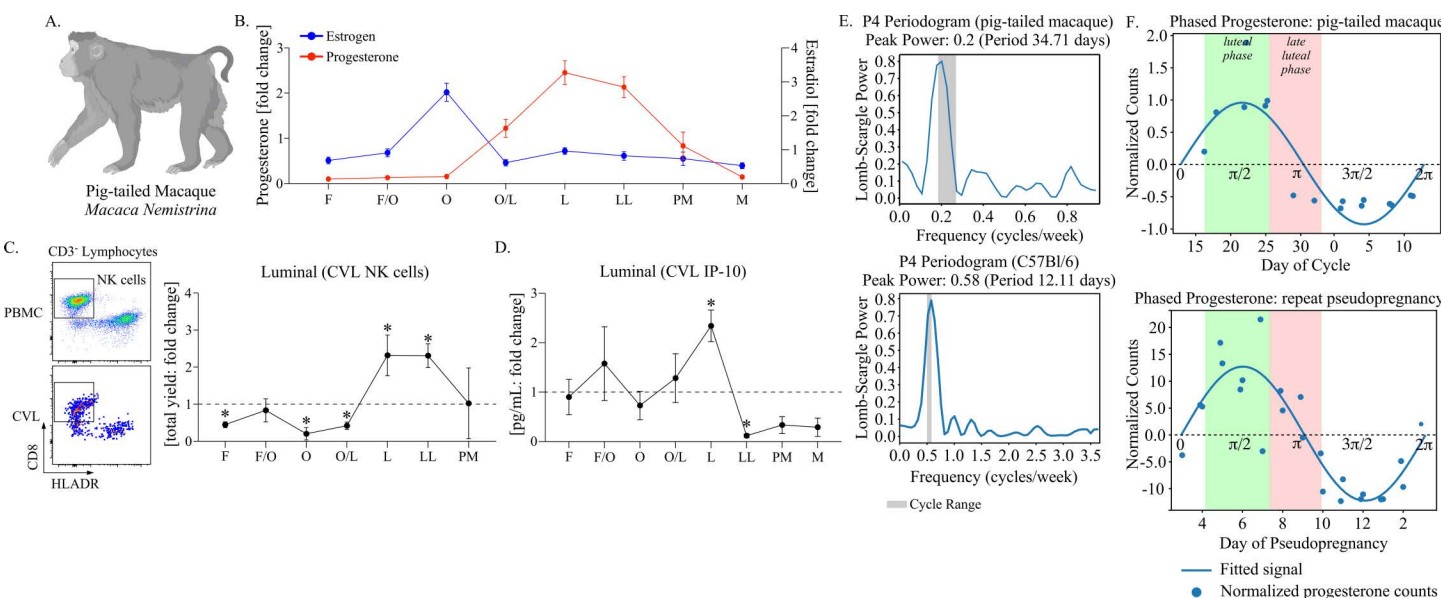

**Fig 3. NK cell oscillations at the cervicovaginal barrier occur under menstrual cycle regulation in pig-tailed macaques. (A).** Cartoon of a pig-tailed macaque (*Macaca Nemistrina*). Figure created using BioRender.com. **(B).** A symbol line graph with SEM depicting the fold change in plasma levels of progesterone (red symbols and lines) and Estrogen (Estradiol, blue symbols and lines) was measured longitudinally from 6 animals and stratified by cycle phase. **(C).** (Left panel) Flow dot plots depicting the strategy for measuring NK cells from PBMC (top) and CVL (bottom). Live, singlet CD45-expressing lymphocytes are first discriminated from granulocytes, myeloid cells, T cells, and B cells and then measured for HLADR negative and CD8 positive populations. (Right panel) a symbol line graph with SEM depicting the fold change in CVL NK cells measured longitudinally and stratified by cycle phase. **(D).** A symbol line graph with SEM depicting the fold change in IP-10 measured from CVL supernatant **(B-D).** The cycle phases are identified as follows: follicular phase (F), follicular/ovulation transition (F/O), ovulation (O), ovulation/luteal transition (O/L), luteal phase (L), late luteal phase (LL), pre-menstruation (PM), and menstruation (M). **(C, D).** Models used to evaluate fold change (against a value of 1) were fit using Wilcoxon rank sum tests. Differences with p-values ≤ 0.05 are indicated by an asterisk. **(E).** Lomb-Scargle periodograms of progesterone with detectable oscillation frequencies shaded in gray and plotted by Lomb-Scargle power of spectral density from a representative pig-tailed macaque (top panel) and a representative mouse undergoing repeat pseudopregnancy (bottom panel). Significance was determined using false alarm probability (1.20x10$^{-5}$ for mouse data) **(F).** Waveforms of progesterone from each animal are depicted to demonstrate phase mapping over an oscillation. Using the oscillation frequency, the progesterone sine wave is estimated using mean normalized counts with the corresponding days of the menstrual cycle or pseudopregnancy cycle plotted on the x-axis. The amplitudes/maximal displacements of the sine waves are identified in radians: π/2 (peak) and 3π/2 (trough). The range of specific menstrual phases from the pig-tailed macaque menstrual cycle is labeled: luteal phase (green shading) and late luteal phase (red shading) with inferred phases indicated in the pseudopregnancy model (n = 3 animals tested).

To assess how the timing of these immune changes relates to the pseudopregnancy model, we compared progesterone kinetics in pig-tailed macaques with mice undergoing repeat pseudopregnancy cycles using periodic analysis with waveform reconstruction (**Fig 3E**). First, we estimated progesterone oscillation frequencies over time as previously described [46]. This approach showed that the average dominant frequency in pig-tailed macaques was 31.2 days (range 28.1-34.71 days), consistent with previous reports [47,48]. In mice undergoing repeat pseudopregnancy, the average dominant frequency was between 10–12 days (range 10.48-12.11 estimated from 3 mice), consistent with our measurements over one cycle (shown in **Fig 1B**). Using these frequencies, we reconstructed waveforms to compare the menstrual cycle and pseudopregnancy (**Fig 3F**). From the pig-tailed macaques, we identified the luteal phase (range 16–25 days) between 60 and 120° (π/3–2π/3) of the rotation and the late luteal phase (range 25–33 days) between 120 and 210° (2π/3–7π/6). Then, by inferring these measurements in the pseudopregnant mice, we observed similar characteristics (n = 3 mice tested). Specifically, the luteal phase mapped to days 3–7 of pseudopregnancy and the late luteal phase to days 7–9. Thus, the time points of pseudopregnancy associated with increased LFRT NK cell infiltration (day 6 and day 8) and IFNγ signaling (day 6) corresponded to the luteal and late luteal phases in the pig-tailed macaques when similar immune changes were detected. Taken together, these data show that pig-tailed macaques exhibit cyclical patterns in NK cell recruitment and IFNγ signaling within the cervicovaginal tissue environment during the luteal and late luteal phase, which is reciprocated in the mouse pseudopregnancy model.

To investigate whether different FRT immune states over pseudopregnancy influence the risk of chlamydial infection, we vaginally challenged mice with $1 \times 10^5$ inclusion forming units (IFU) of *C. muridarum* (**Fig 4A**) and measured bacterial replication at time points corresponding to early infection (day of infection, DOI 3), peak infection (DOI 7), and clearance (DOI 14–28) as previously described [49]. This approach showed that *C. muridarum* challenge at day 8 of pseudopregnancy resulted in a significant reduction in bacterial replication on DOI 3, 7, and 14 as compared with control MPA-treated mice (**Fig 4B**). Day 8 challenge also resulted in lower bacterial replication as compared with day 6 challenge at DOI 3. Additionally, day 8 challenge exhibited reduced *C. muridarum* DNA at DOI 28 as compared with challenge on day 10. Day 4 and day 6 challenges presented mean trends that suggested lower bacterial burden as compared with MPA treatment at the peak of infection, although this was not significant. The only significant difference in day 4 and day 6 challenges was reduced *C. muridarum* DNA at DOI 14 following day 4 challenge as compared with MPA-treated mice. In contrast, *C. muridarum* challenge during the time point of menses onset (administered prior to the detection of menses), which is also when endometrial repair initiates (day 10), resulted in a 1-log increase at peak infection compared to MPA, although this difference was not significant (**Fig 4B**). Due to the increase in *C. muridarum* DNA detection at DOI 28 between day 8 and day 10 challenges, we further tested for infectivity during time points of bacterial clearance (**Fig 4C**) [50]. By measuring the IFU collected from vaginal swabs on the 21st and 28th day of infection (DOI), we found that day 10 challenges yielded greater infectious bacteria compared with day 8 challenges; however, pathologic examination of the FRT at 45 days post-infection showed no evidence of tissue damage, including hydrosalpinx or hydrometra development, which suggested that infections from challenge on day 10 of pseudopregnancy were ultimately cleared [51] (**S5 Fig**). Overall, this approach showed that the timing of challenge over pseudopregnancy resulted in differences in the outcome of *C. muridarum* infection, typified by an early and sustained reduction in bacterial burden following challenge on day 8 of pseudopregnancy.

To better understand how the immune state at day 8 of pseudopregnancy contributed to the differences observed with *C. muridarum* infection following vaginal challenge, we performed immune profiling early during the infection course, 3 days post-challenge, in the cervicovaginal tissues and lumen (**Fig 5**). First, we found that challenge at day 8 of pseudopregnancy was associated with higher levels of multiple proinflammatory cytokines and chemokines compared to animals challenged at day 10 (**Fig 5A**). The strongest differences were IP-10 and IL-5, although significant increases in IL1β, CXCL1, IFNγ, IL-6, and IL27p28 were also detected; all of these cytokines and chemokines identified as elevated early following challenge were also shown in previous studies to correlate with protection against *C. muridarum* [52–56].

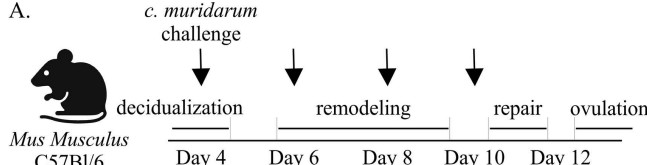

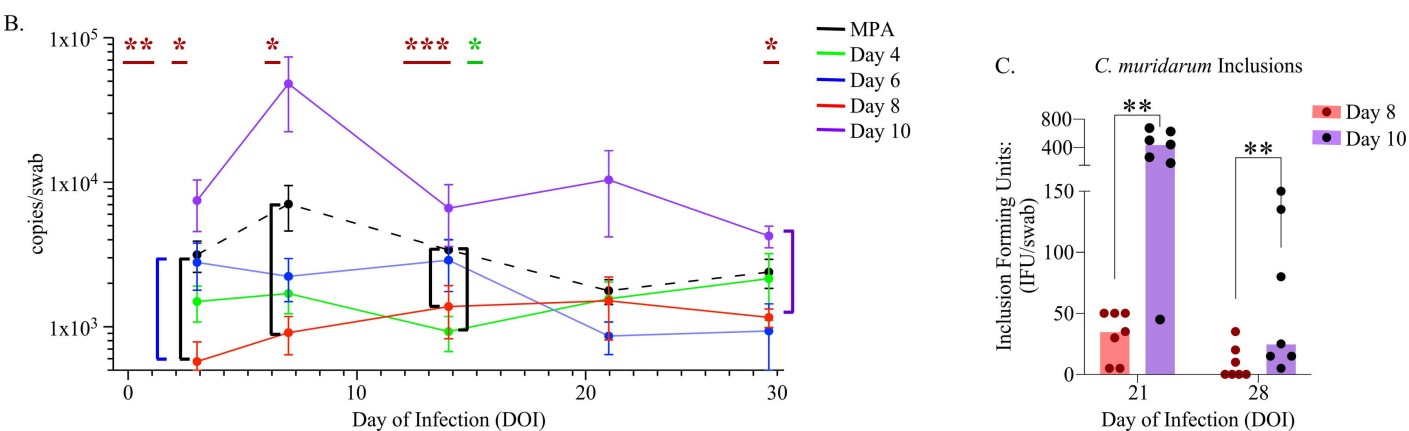

**Fig 4. The timing of *C. muridarum* challenge over pseudopregnancy leads to distinct infection outcomes. (A).** Schematic depiction of the vaginal *C. muridarum* challenge approach at time points of pseudopregnancy (indicated by arrows). Created using BioRender.com. **(B).** Line graphs depicting the mean bacterial burden with SEM over the course of infection determined by PCR and compared with mice administered MPA prior to challenge (n = 10, black dotted lines) at the indicated day of infection (DOI). The graphed data points based on the day of pseudopregnancy at challenge are comprised of 2 separate experiments for each group. The time points of pseudopregnancy at challenge: day 4 (n = 8, green line), day 6 challenge (n = 12, blue line), day 8 challenge (n = 10, red line), and day 10 (n = 14, purple line) are indicated. The colored asterisks at the top of the graph indicates a significant reduction detected from challenges on day 8 (red) and day 4 (green) of pseudopregnancy over the infection, and the corresponding brackets indicates comparisons with day 6 (blue bracket), MPA (black brackets), and D10 (purple bracket). **(C).** A bar graph with symbols (n-values) depicting the mean *C. muridarum* inclusion forming units (IFU) measured per vaginal swab collected at the indicated DOI from challenges occurring at day 8 (n = 7) or day 10 (n = 7) of pseudopregnancy **(B-C).** Models used to compare a difference of means were fit using a mixed-effect model with Geiser-Greenhouse correction: *$p \leq 0.05$, **$p < 0.01$, ***$p < 0.001$.

Next, we evaluated local innate immune cells at these time points by comparing luminal and LFRT neutrophil, macrophage, and NK cell populations (**Fig 5B**). These data showed that the number of FRT leukocytes were increased in mice challenged on day 8 of pseudopregnancy as compared with day 10. Specifically, the greatest yields and mean differences in total leukocytes, neutrophils, macrophage, and NK cells were observed at the luminal surface. These immune populations were also increased in LFRT tissues following challenge at day 8 as compared with day 10, with the exception of macrophage populations, which were not significantly changed. To distinguish changes in innate cell populations that were driven by infection rather than pseudopregnancy, we normalized cell yields to uninfected matched baseline measurements and compared the fold change (**Fig 5C and 5D**). This approach showed that challenge on day 8 resulted in increased luminal and LFRT NK cell populations as compared with day 10. Increased macrophages were also detected following day 8 challenge, although this was only significant in the LFRT. To compare effector responses within the day 8 or day 10 cohorts, we measured fold deviations using Wilcoxon rank sum tests (**Fig 5E and 5F**). This showed that day 8 challenge presented with an overall increase in most cell populations at the luminal surface (leukocytes macrophages and NK cells), while at the LFRT, only NK cells were significantly increased. In contrast, innate immune cell populations following day 10 challenge were more likely to be decreased from their respective baseline values, with significant reductions detected

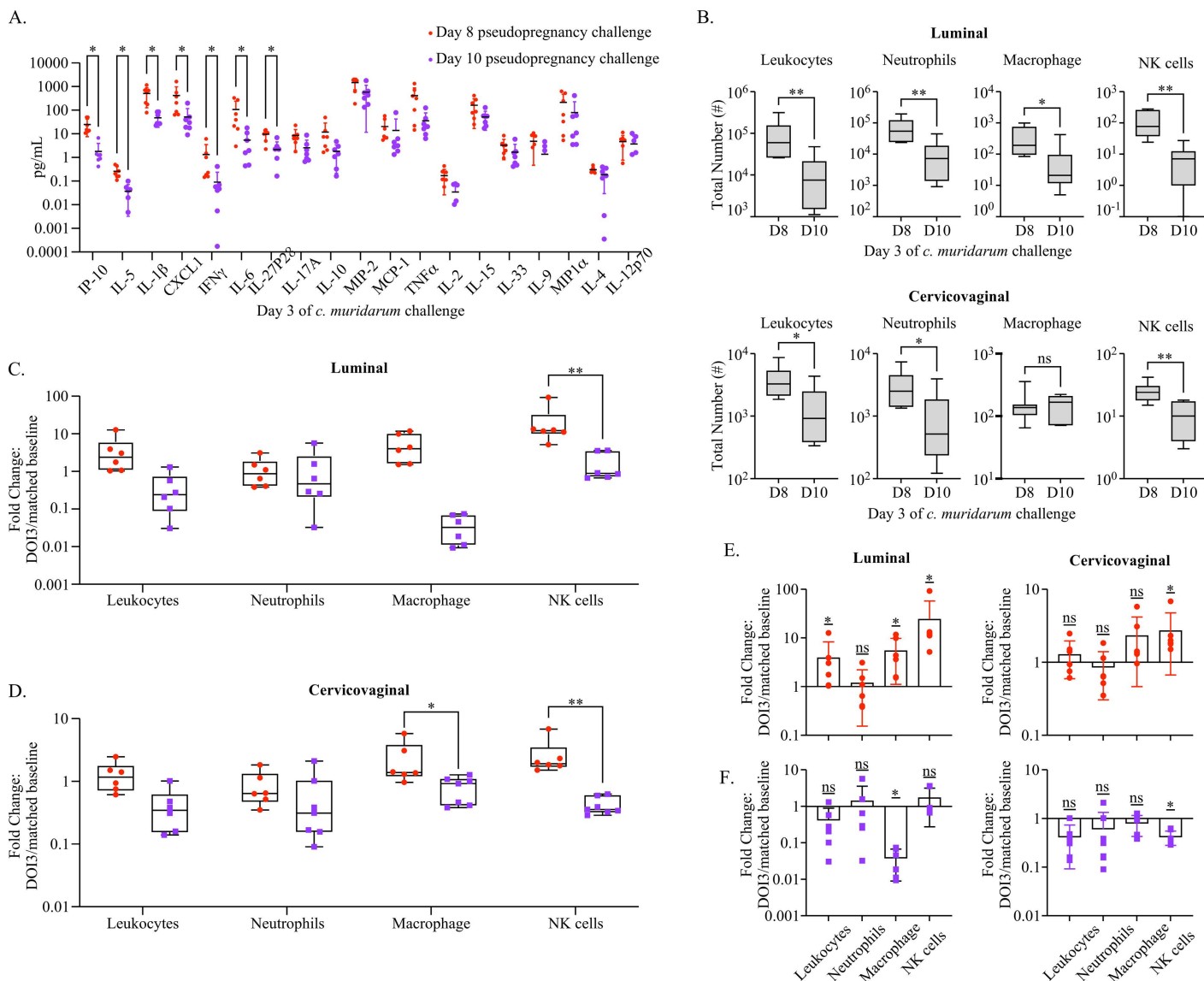

**Fig 5. An early innate effector response is detected following challenges during progesterone withdrawal. (A).** A dot plot graph with the mean and standard deviation (SD) depicting the concentration of indicated proinflammatory cytokine and chemokines measured from CVL supernatant at day 3 of *C. muridarum* infection following challenge at D8 (red) or D10 (purple) of pseudopregnancy. Models used to compare a difference of means were fit using Multiple Mann-Whitney tests and ordered by rank: p-values ≤0.05 are indicated by an asterisk. **(B).** Box and whiskers graphs comparing the total yield of indicated IV-negative innate cell populations from CVL (labeled luminal, top panels) and cervicovaginal tissues (bottom panels) collected on day 3 of *C. muridarum* infection following challenge at day 8 (D8) or day 10 (D10) of pseudopregnancy. Models used to compare a difference of means were fit using unpaired t-tests. **(C).** Box and whiskers graph depicting the fold change in indicated IV-negative innate cell populations at DOI 3 normalized to matched uninfected controls from CVL (baseline n = 7) and **(D).** cervicovaginal tissues (baseline n = 6). **(C, D).** Models used to compare a difference of means were fit using multiple comparisons FDR testing: p-values ≤0.05 are shown. **(E).** Bar graphs with aligned dot plots and SD depicting the fold change in normalized luminal and cervicovaginal innate immune cells at DOI 3 following challenge at day 8 or **(F)**. day 10 of pseudopregnancy. **(E, F).** Models used to evaluate fold change (against a value of 1) were fit using Wilcoxon rank sum tests. **(A-D).** *p ≤ 0.05, **p < 0.01, ***p < 0.001, ****p < 0.0001, ns-not significant.

in luminal macrophages and LFRT NK cells. Taken together, this profiling approach identified a rapid and robust immune response at the infection site in mice challenged on day 8 of pseudopregnancy, as compared with a lack of early effector responses from challenge on day 10.

To identify NK cell contributions to immune protection, we first compared transcriptional profiles in uninfected mice on day 8 of pseudopregnancy with uninfected MPA-treated mice (**Fig 6A**) through RNA sequencing. This showed that while NK cells from the spleen exhibited very few differentially expressed genes (DEGs), the FRT NK cells were transcriptionally distinct populations (**Fig 6B**) with over 500 genes upregulated and 700 genes downregulated in the FRT NK cells at day 8 of pseudopregnancy (**Fig 6C**). To better understand these differences, we compared fold changes in DEGs (**Fig**

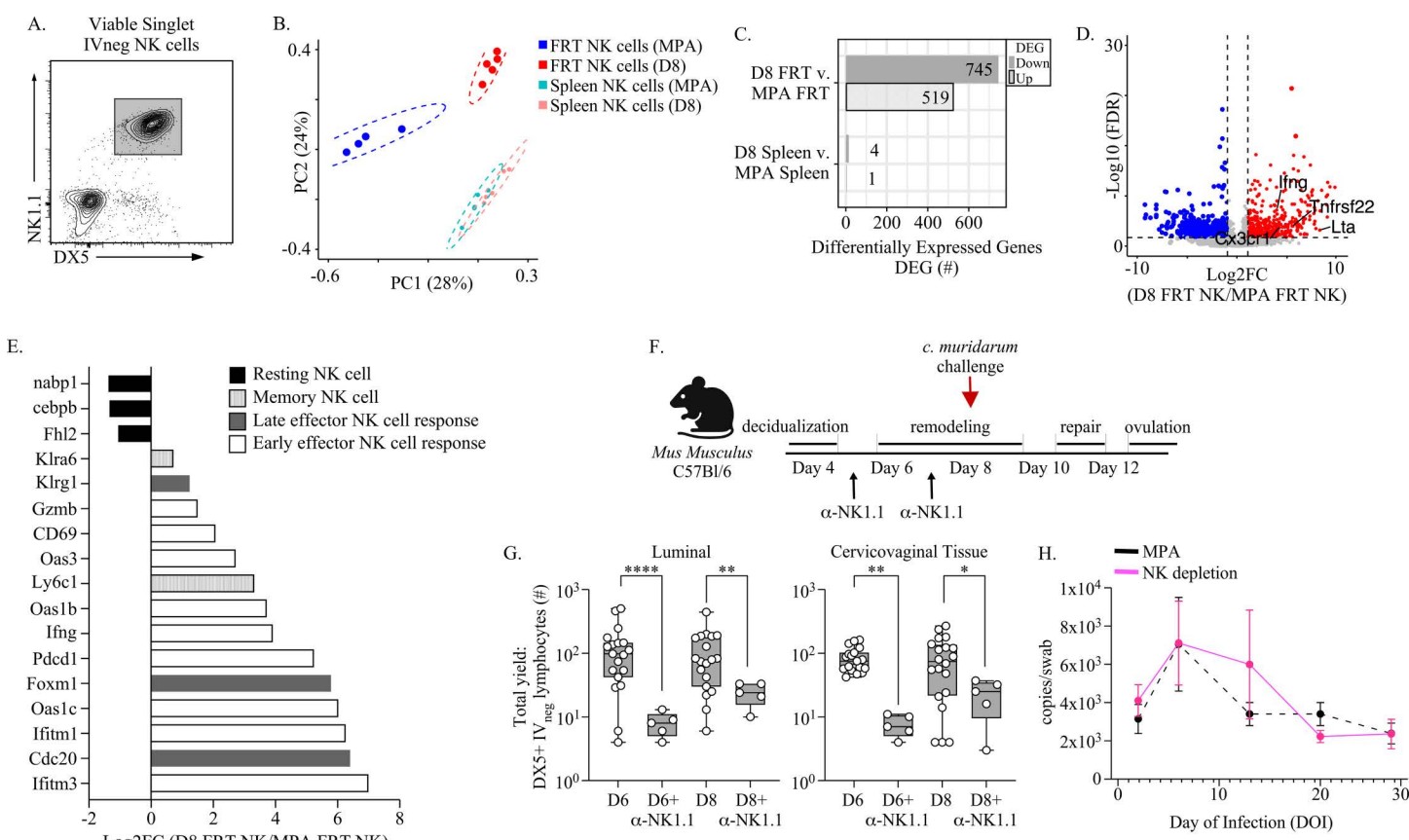

**Fig 6. FRT NK cell baseline states during progesterone withdrawal exhibit immune activation-associated transcriptional programs and are essential for protection against *C. muridarum* challenge. (A).** Sorting strategy for RNA sequencing from NK cells. Viable, singlet lymphocytes were gated from IV negative, CD3, CD19, and Ly6G expressing cells and distinguished by DX5+ and NK1.1+expression. **(B).** PCA plots of detectable genes from indicated cell populations. **(C).** Graph depicting the number of differentially expressed genes (DEGs) when comparing FRT or splenic NK cells. **(D).** A volcano plot showing the log2 fold change in gene expression versus the -log10 of the p-value for FRT NK cell populations at day 8 of pseudopregnancy versus MPA with specific genes of interest that are upregulated in the D8 FRT NK cells labeled. **(E).** Bar graph of DEGs associated with resting or activated NK cells. **(F).** Schematic depicting the approach for depleting NK cells during time points of endometrial remodeling and prior to vaginal *C. muridarum* challenge at day 8 of pseudopregnancy. αNK1.1 antibody injections were administered on day 5 and day 7 of pseudopregnancy. Schematic created using BioRender.com **(G).** NK cells are measured at the indicated time points over pseudopregnancy from the cervicovaginal lumen (left panel) or underlying tissues (right panel) following NK cell depletion and compared with mice not treated with αNK1.1 antibody over pseudopregnancy (originally shown in Fig 2). **(H).** The bacterial burden of *C. muridarum* is measured from vaginal swabs collected over the course of infection from mice that are administered αNK1.1 antibody (n=11, pink lines) over endometrial remodeling and compared with historical measurements from mice treated with MPA prior to challenge (n=10, dotted line originally shown in Fig 4B and 4C). **(G, H).** Models used to compare a difference of means were fit using multiple comparisons with FDR testing: *p≤0.05, **p<0.01, ***p<0.001, ****p<0.0001.

6D), which showed that day 8 FRT NK cells exhibited an increase in gene transcripts associated with NK cell regulation of immune effector functions, including Tnfrsf22, which transcribes a TNFαreceptor that mediates NK cell proliferation and cytotoxic function [57], and CX3CR1, a chemokine receptor which regulates cell trafficking into inflammatory tissue sites [58]. Other notable genes that were increased in FRT NK cells at day 8 included the effector cytokines lymphotoxin-alpha (LT-α) and IFNγ. Because we identified enrichment of gene transcripts involved in immune activation and effector responses, we further explored effector profiles using a previously characterized transcriptomic stratification approach that distinguished gene signatures associated with NK cell activation phases from those in a resting state [59] (Fig 6E). This analysis showed that FRT NK cells from MPA-treated mice were only enriched for resting state properties, while FRT NK cells from day 8 of pseudopregnancy were enriched for early effector responses, although increased transcripts associated with late effector and memory states were also detected.

Finally, to determine whether FRT NK cells, which have previously been identified as important for immune control of genital chlamydia infection [35,60], were also important to the early control of *C. muridarum* on day 8 of pseudopregnancy, we performed antibody-mediated depletion of NK cells over the time points of endometrial remodeling through 2 series of intraperitoneal (IP) injections with αNK1.1 at day 5 and day 7 of pseudopregnancy (Fig 6F). As expected, αNK1.1 administration significantly reduced LFRT and luminal NK cells on day 6 and day 8 of pseudopregnancy (Fig 6G). Next, we vaginally challenged NK cell-depleted mice at day 8 with *C. muridarum* and measured bacterial burden over the infection course (Fig 6H). In contrast to the protection observed at day 8 when NK cells are present (Fig 4B), depletion of NK cells resulted in a productive infection following day 8 challenge, with similar infection kinetics compared to mice administered MPA. Taken together, these data show that the immune events of endometrial remodeling and menstruation drive the recruitment of activated NK cells into the cervicovaginal barrier, which plays an essential role in early defense against *C. muridarum*.

## Discussion

A growing body of work demonstrates that identifying immune correlates of protection is important for developing biomedical interventions that can prevent infections or limit disease burden caused by pathogens, including those that cause STIs [61–67]. Although uncovering the complex relationships that occur between the immune response and an invading pathogen can provide critical information for understanding how to prevent diseases, our insights into the potential roles of the menstrual cycle in determining such correlates are greatly limited. Despite the fact that the immune system is fundamental for menstruation, elucidating how this process can also impact mucosal immune defense against genitourinary pathogens, including *C. trachomatis,* has been obstructed by the lack of accessible animal models. The experimental and genetic tools available for laboratory mice would allow mechanistic investigations into regional immune dynamics occurring throughout the FRT under menstrual cycle regulation and determination of how changes to immune barrier defenses can alter infection outcomes. Here, we employed the murine pseudopregnancy approach for inducing menstruation in the context of a vaginal chlamydia challenge system in order to explore how the process of endometrial remodeling and repair drives spatiotemporal immune changes in the FRT mucosae and directly test how these alterations shape infections by *C. muridarum*. Using this approach, we discovered that over the course of decidualization, endometrial remodeling, and menstruation, the cervicovaginal tissue undergoes substantial immune alterations that mimic the changes occurring in uterine tissues. These changes were characterized by a transient influx of neutrophils, macrophages, and NK cells paired with increased proinflammatory mediators during the endometrial remodeling process.

From our vaginal *C. muridarum* infection approach over pseudopregnancy, we discovered that challenges administered immediately prior to menses onset were comparatively less protective. Interestingly, previous investigations have identified that endometrial repair begins at the very start of decidual shedding and is typified by an immune shift towards more immunosuppressive properties, which are needed to prevent tissue scarring and allow regeneration [68,69]. In this study, we found that challenges occurring during this time point resulted in robust infections and

showed evidence of dampened early effecter responses. These findings suggest that the conditions of uterine repair may be more hospitable for chlamydial infections in cervicovaginal tissues. However, future studies will be needed to comprehensively examine this time span in the murine menstruation model in order to better identify and understand those potential risk factors.

Contrary to conditions of uterine repair, we discovered that *C. muridarum* challenges during progesterone withdrawal resulted in rapid immune responses with an early and sustained reduction in bacterial burdens. We further identified through progesterone waveform reconstruction that this time point corresponded to the late luteal phase of the menstrual cycle and was defined by NK cell enrichment at the luminal surface and underlying cervicovaginal tissues during a time when FRT tissue-localized leukocyte populations were contracting. Interestingly, through RNA sequencing, we were able to identify that these FRT NK cells were a unique population and were enriched for programming indicative of an immune-activated state. Previous studies have identified that both *C. trachomatis* and *C. muridarum* are susceptible to direct NK cell-mediated killing of host cells. Furthermore, NK cells have been shown to enhance the activation of Th1 cells, which can provide effective primary and secondary responses against genital chlamydia infections [3,28,35,70,71]. Therefore, to directly test the ability of these cells to protect against chlamydial infection, we depleted NK cells in mice during progesterone withdrawal prior to *C. muridarum* challenge. These data showed that NK cell-specific depletion at this time point resulted in productive infection, similar to the level detected in mice administered the hormonal contraceptive MPA. These findings demonstrate that the menstrual cycle regulates cyclical localization and enrichment of NK cell populations at the cervicovaginal barrier, where they can provide rapid immune protection against primary chlamydial infections.

The baseline "activation" states of both the innate and adaptive arms of the immune system have been previously shown to predict vaccine efficacies and the likelihood of disease development [46,67,72,73]. For example, recent work has identified that basal states of innate immune cells, including NK cell populations, can indicate greater protection against disease development following influenza infection [67]. While many host-intrinsic properties such as age, sex, health, and genetics can influence immune homeostasis, the specific contributions of the menstrual cycle to these baseline immune states in the context of disease risk and vaccine efficacies are far less characterized. Given the role of NK cells in activating adaptive immune responses, including T cells, it is tempting to speculate that the oscillating proinflammatory signals occurring over the menstrual cycle, systemically and within the FRT [46,48], might influence both protection from infection and the establishment of immune memory. Thus, this possibility should be further explored in the context of the menstrual cycle.

To conclude, this study demonstrates that the process of menstruation regulates regional immune states throughout the upper and lower FRT, which can impact mucosal barrier defenses against chlamydial infections. We further demonstrate that the murine pseudopregnancy approach for inducing menstruation is a valuable tool for investigating how this process drives mucosal immune dynamics in the FRT, and we posit this approach will be beneficial in the development of novel biomedical strategies that can strengthen immunity against genitourinary pathogens.

## Materials and methods

### Ethics statement

All experiments conducted on animals were in accordance with the approved protocols. Animal studies and experimental protocols were approved by the Emory University Institutional Animal Care and Use Committee (IACUC).

### Mice

C57BL/6J (wild-type) mice and Swiss Webster outbred mice (Taconic Biosciences) were housed under specific Animal Biosafety Level 2 conditions at Emory University. Pseudopregnancy: 6–12 week old female C57BL/6J were mated

with Swiss Webster vasectomized males to induce pseudopregnancy. At 4 days post-mating, female mice received an intrauterine injection of sesame seed oil (Sigma-Aldrich) using mNSET devices (Paratechs) as previously described [23]. Cycle phase kinetics were determined using vaginal cytology via Hemotoxin and Eosin staining (H&E), in addition to visible detection of menstruation. For controls, a subset of mice (not undergoing pseudopregnancy) received a subcutaneous injection with 3mg of Medroxyprogesterone acetate (MPA, Prasco) 2 weeks prior to necropsies. NK cell depletion: To deplete NK cells in vivo, mice were intraperitoneally (IP) injected with 200ug α-NK1.1 (clone PK136, BioXcell).

### *C. muridarum* challenge

*Chlamydia muridarum (C. muridarum)* Mouse Pneumonitis Nigg II strain (ATCC) was cultured in HeLa cells and purified by density centrifugation as previously described [49]. Aliquots were stored in sodium phosphate glutamate buffer (SPG) at −80°C. The inclusion forming units (IFU) from purified elementary bodies were determined by infection of HeLa 229 cells and enumeration of inclusions by microscopy. For vaginal infection, $10^5$ IFU of *C. muridarum* in SPG buffer was deposited into the vaginal vault as previously described [49]. To measure bacterial burden, DNA was purified (Qiagen) from vaginal swabs (Puritan) and quantified by PCR. To measure IFU over infection, vaginal swabs were collected into SPG buffer containing glass beads and vortexed for one minute before adding the supernatant to confluent Hela cell cultures. Following low-speed centrifugation for 40 minutes, SPG buffer was replaced with infection media and cultured for 24 hours, as previously described [74]. Inclusions were distinguished by Giemsa stain (Sigma-Aldrich) and enumerated using a light microscope.

### PCR

The *C. muridarum* bacterial burden was measured using Droplet Digital PCR (ddPCR) technology (Bio-Rad) according to manufacturer recommendations [75–77] and was first validated for bacterial burden using *C. muridarum* standards (S6 Fig). In brief, a mixture containing 2x QX200 ddPC EVAgreen supermix, mixed 16SR (chlamydia muridarum) forward (AGTCTGCAACTCGACTAC) and reverse (GGCTACCTTGTTACGACT) primers (4µM), ultrapure water, and the DNA sample was used to amplify a fragment of the gene of interest. 20µL of this mixture was added to 70µL of droplet generation oil, and after the droplet generation step, the suspension was used to perform ddPCR in a 96-well PCR plate. The fluorescent signal was read by a QX200 Droplet Reader (Bio-Rad) and analyzed with QuantaSoft software. The gating for positive droplets was set according to the positive and negative controls read with each plate.

### Murine tissue processing

Intravital vascular staining: Intravital vascular (IV) staining in mice was performed prior to euthanasia and tissue harvest as previously described [16]. In brief, to discriminate immune cells resident in various tissues from those in circulation, 1.5µg fluorophore-conjugated anti-CD45 antibody in 200µl 1xPBS was intravenously injected into the tail vein of mice and allowed to circulate for at least 15 minutes; post-injection, mice were euthanized with Avertin (2,2,2-tribromoethanol; Sigma-Aldrich) and exsanguinated prior to CVL and tissue collection. CVL collection: To collect and compare cervico-vaginal luminal cells in mice, 50ul of sterile PBS was deposited and retracted into the vaginal vault at equal repetitions lasting about 30 seconds. Tissue Processing: FRT tissues were digested using collagenase type II (62.5 U/ml) and DNase I (0.083 U/ml) (STEMCELL Technologies). Cell suspensions were separated by Percoll (GE healthcare life sciences) discontinuous density centrifugation. Enriched leukocytes were washed and resuspended in cell media for phenotyping. For measurement of sex hormones from mice, blood was collected at necropsy by cardiac puncture, or in the case of longitudinal monitoring over repeat pseudopregnancy; a few drops were collected by tail vein lancet into 1.3mL EDTA blood tubes (Fisher Scientific) and then centrifuged for plasma collection.

## Tissue pathology scoring

Histopathologic analysis was performed on murine FRT tissues 45 days post-infection, as previously described [78]. Longitudinal 4 micrometer (µm) sections from formalin-fixed FRT tissues were stained using H&E and evaluated by a pathologist. Neutrophils, plasma cells, mononuclear cells, fibrosis, and oviduct dilation were measured from the vaginal vault, cervix, uterine horns, oviducts, and mesosalpinx. A score of 0–1 was considered normal to trace levels, a score of 2–3 indicated mild to moderate detection, and a score of 4 indicated severe infiltration or tissue damage.

## NHP

For this study, blood and CVL were collected from 6 healthy female pig-tailed macaques of reproductive age over a period of 9 weeks. All NHP procedures were first approved by the CDC Institutional Animal Care and Use Committee. Macaques were housed at the CDC under the full care of CDC veterinarians in accordance with the standards incorporated in the *Guide for the Care and Use of Laboratory Animals* (National Research Council of the National Academies, 2010). All procedures were performed under anesthesia using ketamine, and all efforts were made to minimize suffering, improve housing conditions, and provide enrichment opportunities. 5mL of blood was collected in 8 mL sodium citrate-containing CPT tubes (BD Biosciences) and separated into plasma and PBMC by centrifugation. CVL specimens (10 mL collections) were processed as previously described [46,48].

## Sex-hormone measurement and estimating menstrual cycle phase

Progesterone [P4] and Estradiol [E2] levels in plasma were quantified by immunoassay in one single batch per species. Assay services were provided by the Biomarkers Core Laboratory at the Yerkes National Primate Research Center. The menstrual cycle phase of pig-tailed macaques was estimated by P4 and E2 kinetics relative to a 32-day menstrual cycle (average length of pigtail macaque menstrual cycle) and by observed menstruation.

## Soluble cytokine/chemokine measurement

CVL supernatant and blood plasma were measured and analyzed for cytokine/chemokines through the Emory Multiplexed Immunoassay Core using the Meso Scale Discovery (MSD) platform using a murine and NHP multiplex assay kit in one batch run per species.

## Microscopy

Tissues were flash-frozen in Tissue-Tek Optimal Cutting Temperature (O.C.T.) Compound (Sakura) and stored at -80°. Tissue blocks were sectioned onto microslides at 6–8 micron thickness using a cryostat and stored at -80° prior to staining. For immunofluorescent staining, slides were first fixed in 75:25 acetone/ethanol and then blocked with donkey and rabbit serum. The following antibodies were used for staining: rat anti-E-cadherin (Abcam clone: DECMA-1) paired with Alexa Flour 488 anti-rat IgG (Abcam), Alexa Flour 594 anti-Ly6G/Ly6C (Biolegend clone: Gr-1), Alexa Flour 647 anti-CD31 (Abcam clone: MEC 7.46). Following antibody staining, mounting medium with DAPI (Abcam) was added to slides prior to microscopy. Imaging was performed on a Zeiss Axio Observer Z1 with an Axiocam 506 monochromatic camera. Image processing was performed using Zen 2 software.

## Flow cytometry

Single-cell suspensions were first stained for viability using Zombie NIR Fixable Viability Kits (Biolegend), followed by cell surface staining and measurements using a BD LSRFortessa or LSRII high-parameter cell analyzer, and flow data was acquired using FACS DIVA software (BD Biosciences). Data was analyzed using FlowJo software (TreeStar, Inc.). The following fluorochrome-conjugated antibodies were used:

## Murine antibodies

| Marker | Clone | Channel | Company |
|---|---|---|---|
| IV-CD45 | 30-F11 | BD Horizon PE-CF594 | BD Biosciences |
| CD45 | 30-F11 | BD Pharmingen FITC | BD Biosciences |
| CD49b | DX5 | Phycoerythrin (PE) | BioLegend |
| CD68 | FA-11 | PerCP/Cyanine5.5 | BioLegend |
| F4/80 | BM8 | Brilliant Violet 650 | BioLegend |
| CD11b | M1/70 | BD Horizon BUV395 | BD Biosciences |
| MHCII | M5/114.15.2 | PE/Cyanine7 | BioLegend |
| Ly6G | 1A8 | Brilliant Violet 510 | BioLegend |
| Ly6G | 1A8 | Alexa Fluor 700 | BioLegend |
| CD115 | T38-320 | BD OptiBuild BUV496 | BD Biosciences |
| NK1.1 | S17016D | Allophycocyanin (APC) | BioLegend |
| CD3 | 17A2 | Alexa Fluor 700 | BioLegend |
| CD19 | 1D3/CD19 | Alexa Fluor 700 | BioLegend |

## NHP antibodies

| Marker | Clone | Channel | Company |
|---|---|---|---|
| CD45 | D058-1283 | BV421 | BD Biosciences |
| CD3 | SP34–2 | Alexa Fluor 700 | BD Biosciences |
| CD8 | RPA-T8 | Brilliant Violet 510 | BioLegend |
| CD10 | HI10a | Allophycocyanin (APC) | BioLegend |
| CD14 | M5E2 | Brilliant Violet 570 | BioLegend |
| CD20 | 2H7 | BD Pharmingen FITC | BD Biosciences |
| HLADR | L243 | Brilliant Violet 605 | BioLegend |

## RNA sequencing

Cell suspensions from total FRT (excluding oviducts and ovaries) and spleen were enriched for NK cells using NK cell isolation kits (Miltenyi Biotec) and stained for viability and cell surface characteristics to distinguish tissue localized populations (IVneg, CD45+DX5+, NK1.1+lymphocytes that were negative for IV+, Ly6G, CD3, and CD19) for mechanical sorting of 100–1000 cells into RLT buffer (Qiagen) supplemented with 1% βmercaptoethanol using a Cytek Aurora CS System. Total RNA was purified using the Quick-RNA Microprep kit (Zymo Research). All resulting RNA was used as an input for complementary DNA synthesis using the SMART-Seq v4 kit (Takara Bio) and 12 cycles of PCR amplification. 200 picograms of cDNA were converted to a sequencing library using the NexteraXT DNA Library Prep Kit and NexteraXT indexing primers (Illumina) with 12 additional cycles of PCR. Final libraries were pooled at equimolar ratios and sequenced on a HiSeq2500 using 50-bp paired-end sequencing or a NextSeq500 using 75-bp paired-end sequencing. Raw fastq files were mapped to the mm10 build of the mouse genome using STAR [79] with the GENCODE v17 reference transcriptome. The overlap of reads with exons was computed and summarized with the GenomicRanges package [80], and data normalized to fragments per kilobase per million (FPKM). Genes that were expressed at a minimum of three reads per million (RPM) in all samples were considered expressed. Differentially expressed genes (DEGs) were determined using the glm function in DESeq2 [81] using the mouse from which each cell type originated as a covariate. Genes with a false discovery rate (FDR) < 0.05 and absolute log2(FC) > 1 were considered to be significant. All code for data processing and display is available upon request.

## Periodogram and waveform reconstruction

The Lomb-Scargle method was used to estimate the strength of dominant frequencies in blood plasma progesterone concentrations over a time series [82,83]. The Lomb-Scargle method is a specialized application of least-squares spectral analysis that uses the form of a sinusoidal wave function:

$$\phi(t) = A\sin(2\pi ft) + B\cos(2\pi ft) + C,$$

Where frequency is given by f and time is given by T, to fit coefficients A, B, C such that the residual differences between the mean normalized data and fitted sinusoid is minimized by least-squares regression. We computed generalized Lomb-Scargle periodograms that account for data with a non-zero mean [84]. False alarm probability is then calculated for a resonant frequency of interest, signifying the likelihood of observing this level of power in a random system. Using the strongest resonant frequency, we reconstructed the sinusoid with normalized data points for cycle fitting. Three mice induced to undergo repeat pseudopregnancy were used in this analysis. Astropy software was used for periodogram and waveform reconstruction.

## Statistical testing

Statistical analysis was performed using Prism (GraphPad software). Each figure legend indicates the methods of comparisons and corrections.

## Supporting information

**S1 Fig.  Images from** Fig 2 **with DAPI staining added (A).** Fluorescence microscopy images taken at 20x magnification identify blood vessels (CD31 expression in red), myeloid cells (Gr-1 expression in purple), epithelial cells (EPCAM expression in green), and DAPI staining (cell nuclei blue) in the lower FRT (LFRT top panels), and uterine horns (bottom panels) at day 4, **(B).** day 6, and **(C).** day 8 of pseudopregnancy. **(D).** Additional images of the LFRT on day 8. (A-C). The white scale bars indicate 100 μm length.
(TIF)

**S2 Fig.  The total leukocyte yield from indicated FRT tissue sites are plotted as bar and whiskers graphs over pseudopregnancy and compared with mice administered MPA or mice administered sesame seed oil in the absence of pseudopregnancy as a control.** Models used to compare a difference of means were fit using multiple comparisons with FDR testing: *p ≤ 0.05, **p < 0.01, ***p < 0.001, ****p < 0.0001.
(TIF)

**S3 Fig.  The total yield of (A). leukocytes, (B).** Neutrophils, (C). Macrophage, and (D). NK cells from indicated FRT tissues are plotted as box and whiskers graphs and compared by the first (taken from Figs 2 and S2) or second (white with black squares representing individual values labeled as a repeat) pseudopregnancy cycle. Models used to compare a difference of means were fit using multiple comparisons with FDR testing: *p ≤ 0.05, **p < 0.01, ***p < 0.001, ****p < 0.0001. ns-not significant.
(TIF)

**S4 Fig.  Box and whiskers graphs depicting the percent frequency of NK cells from the total leukocyte population (shown in** S2 Fig**) from (A).** CVL or (B). LFRT at indicated days of pseudopregnancy or in MPA-treated mice. Models used to compare a difference of means were fit using multiple comparisons with FDR testing: **p < 0.01, ***p < 0.001, ****p < 0.0001.
(TIF)

**S5 Fig. A symbol graph of FRT tissue scoring from mice 45 days post vaginal challenge with 1x10$^5$ IFU of *C. muridarum* stratified by the time of challenge.** Groups: day 8 of pseudopregnancy (n=7), day 10 of pseudopregnancy (n=7), MPA (n=10), and control mice (naïve n=2). Tissues scoring 0–1 identify little to no detection, 2–3 identify moderate detection, and 4 identify severe infiltration or damage. No fibrosis or tissue damage, including oviduct dilation, was detected. Models used to evaluate score deviation (against a value of 1) were fit using Wilcoxon rank sum tests. No group scores were significantly increased.
(TIF)

**S6 Fig. An XY graph with prediction bands plotting the ddPCR quantification using dilutions taken from DNA extracted from 1x10$^5$ IFU of *C. muridarum*.** Distributions were tested by Spearman's correlations.
(TIF)

**S1 Table. The mean levels of indicated cytokines and chemokines (pg/mL) with the SEM over pseudopregnancy and in mice administered MPA.**
(DOCX)

## Acknowledgments

We thank Dr. Marion Rudolph at Bayer HealthCare for helpful insights into the pseudopregnancy approach. From the CDC Division of HIV Prevention (DHP), we thank Dr. J. Gerardo-Garcia Lerma, Dr. Jim Smith, Sunita Sharma, Susan Rhone, James Mitchell, and Frank Deyonks for their assistance in the NHP studies. From the Emory University Medical School Pediatric/Winship Flow Cytometry Core, we thank Dr. Igor Albizua-Santin for helpful insights into our strategy for NK cell sorting. From Emory University Medical School, we thank Dr. Jacob E. Kohlmeier for assistance with mouse studies. Assay services were provided by the Biomarkers Core Laboratory at the Emory National Primate Research Center.

## Author contributions

**Conceptualization:** Laurel A. Lawrence, Mark Elliott Williams, Zheng-Rong Tiger Li, Melissa A. Roy, Alison Kohlmeier.

**Data curation:** Mark Elliott Williams, Paola Vidal, Richa Varughese, Zheng-Rong Tiger Li, Melissa A. Roy, Steven C Tuske, Alison Kohlmeier.

**Formal analysis:** Mark Elliott Williams, Paola Vidal, Richa Varughese, Zheng-Rong Tiger Li, Melissa A. Roy, Steven C Tuske, Alison Kohlmeier.

**Funding acquisition:** Alison Kohlmeier.

**Investigation:** Laurel A. Lawrence, Mark Elliott Williams, Paola Vidal, Richa Varughese, Thien Duy Chen, Steven C Tuske, Alison Kohlmeier.

**Methodology:** Laurel A. Lawrence, Mark Elliott Williams, Richa Varughese, Zheng-Rong Tiger Li, Alison Kohlmeier.

**Project administration:** Laurel A. Lawrence, Mark Elliott Williams, Anice C. Lowen, Christopher D. Scharer, Alison Kohlmeier.

**Resources:** Anice C. Lowen, Christopher D. Scharer, Alison Kohlmeier.

**Software:** Mark Elliott Williams.

**Supervision:** Anice C. Lowen, Christopher D. Scharer, William M. Shafer, Alison Kohlmeier.

**Validation:** Mark Elliott Williams, Zheng-Rong Tiger Li, Thien Duy Chen, Melissa A. Roy, Alison Kohlmeier.

**Visualization:** Mark Elliott Williams, Paola Vidal, Richa Varughese, Steven C Tuske, Alison Kohlmeier.

**Writing – original draft:** Mark Elliott Williams, Paola Vidal, Richa Varughese, Steven C Tuske, Alison Kohlmeier.

**Writing – review & editing:** Anice C. Lowen, Christopher D. Scharer, William M. Shafer, Alison Kohlmeier.

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
