## [Decision Letter · Decision Letter 0]

Dear Dr. Kohlmeier,

Thank you very much for submitting your manuscript "Murine modeling of menstruation identifies immune correlates of protection during Chlamydia muridarum challenge." for consideration at PLOS Pathogens. As with all papers reviewed by the journal, your manuscript was reviewed by members of the editorial board and by several independent reviewers. In light of the reviews (below this email), we would like to invite the resubmission of a significantly-revised version that takes into account the reviewers' comments.

We cannot make any decision about publication until we have seen the revised manuscript and your response to the reviewers' comments. Your revised manuscript is also likely to be sent to reviewers for further evaluation.

Sincerely,

Jorn Coers

Academic Editor

PLOS Pathogens

Matthew Wolfgang

Section Editor

PLOS Pathogens

Michael Malim

Editor-in-Chief

PLOS Pathogens

orcid.org/0000-0002-7699-2064

Reviewer's Responses to Questions

**Part I - Summary**

Reviewer #1: The authors created the pseudopregnancy condition in naïve C57Bl/6 mice (Fig. 1) and monitored inflammatory cell and cytokine distributions (Fig. 2) in the lumen and different genital tissues along the pseudo-menstruation process using the MPA-treated mice as the control. They found that mice were resistant to C. muridarum infection until day 10 after pseudo-menstruation induction (Fig. 4). The increased susceptibility to C. muridarum infection on day 10 correlated with significantly decreased levels of 7 cytokines (IP-10, IL-5, IL-1b, CXCL1, IFNg, IL-6& IL-27p28) and leukocytes including neutrophils, macrophages, and NK cells (Fig. 5, no data on other cell types). Depleting with an anti-NK1.1 antibody significantly reduced NK cells and increased mouse susceptibility to C. muridarum infection on day 8 after induction of pseudo-menstruation (Fig. 6). Thus, the authors claimed that “murine modeling of menstruation identifies immune correlates of protection during C. muridarum challenge”. The main issue with this manuscript is that the same findings have been repeatedly published using MPA-treated mice, which is a much easier model to work with. In addition, some data conflicted with each other, and not all of the data presented in the manuscript supported the authors’ claims.

1. Figure 1 describes the pseudopregnancy model of menstruation. No new finding was presented. Panels F-H seemed to show the highest level of white blood cells circulating to the uterine horn tissues on day 8 after pseudopregnancy induction. However, it is not clear whether this was due to more circulation by the intravenously injected anti-CD45 antibody or the hematogenously labeled CD45+ cells. It is necessary to identify the subtypes of the FITC+ and/or PE+ cells if the authors want to claim that NK cells are among the recruited cells.

2. Figure 2, panel B, what does “iVneg” displayed along the Y-axis mean? Did this mean the detected cells were CD45PE negative or were genital tissue-resident cells?

3. In Figure 2 B-D, it appeared that most cells and cytokines peaked on day 6 after pseudopregnancy induction. Their levels on day 4 were lower than those on day 10. This trend was inconsistent with the data reported in Figure 5 and contradicted their claim that the increased susceptibility to C. muridarum on day 10 correlated with the decreased levels of cytokines and inflammatory cells.

4. Figure 4, based on the C. muridarum burden-shedding time courses presented in panes B&C, productive infection only occurred when C. muridarum was inoculated to MPA-treated mice or the pseudopregnancy-induced mice on day 10 after the induction since these two curves peaked on day 7 after infection. However, all inoculation conditions resulted in long-lasting shedding courses of chlamydial burden without clearance by the end of the observation on day 30 after infection. Obviously, the monitoring is not thorough enough to support unbiased conclusions. The authors must either prolong the observation time or measure viable organisms, which may allow the authors to acquire more comprehensive data to support unbiased conclusions.

5. Why did the authors only compare the cytokine and cell levels between day 8 and day 10 after pseudopregnancy induction? The day 4 and day 6 mice were as resistant to C. muridarum infection as the day 8 mice!!! It is unscientific to pick and choose data to support one’s model.

6. Figure 6, anti-NK1.1 depletion on day 8 after pseudopregnancy induction made the mice develop a shedding course similar to that of the MPA-treated mice. The role of NK1.1+ cells in controlling chlamydial infection in the female genital tract or systemic chlamydial infection has been demonstrated repeatedly in the literature. For example, Tseng and Rank depleted NK cells to increase chlamydial burden in MPA-treated mice (doi: 10.1128/iai.66.12.5867-5875.1998) while Xu et al used NK1.1 depletion to increase chlamydial burden in Rag1 KO mice (DOI: https://doi.org/10.1128/iai.00152-20). Although NK1.1+ cells may contain many different subsets of cells, since the NK1.1 depletion is always accompanied by a decrease in IFNg, the enhanced chlamydial burden has been attributed to the decrease in IFNg.

7. NK cells are considered circulatory cells. The genital tract accessibility to circulatory cells peaked on day 8 after pseudopregnancy induction (Fig. 2). Why did the mice become susceptible to C. muridarum infection on day 10? The true mechanisms may be more complex than NK cells, addressing which may allow the authors to produce real new data.

Reviewer #2: This manuscript by Lawrence at al. addresses a major research gap in the study of sexually transmitted infections, which is how immune responses are altered by the tissue remodeling processes that occur during a menstrual cycle. In light of the high prevalence and morbidity caused by of chlamydial infections in females of reproductive age and clinical observations that suggest reproductive hormones alter susceptibility and perhaps disease severity in women with C. trachomatis (Ct) infection, there is a major lack of fundamental immunological information in this area.

Strengths of this work include i.) the use of well-established pseudopregnancy model in mice to allow mimicking of the menstrual cycle in a murine model that is an accepted surrogate for Ct infection and disease in humans; ii.) the measurement of parameters over the course of the menstrual cycle in pig-tailed macaques to determine whether observations from the pseudo-pregnant mouse model are similar in a nonhuman primate; and iii) direct testing of one of the main hypotheses resulting from these first two studies using NK cell-depleted mice. The major finding that NK cells and IFN-gamma levels are highest during periods of lesser susceptibility and that NK cells play a protective role is important and may also be helpful in vaccine development. There is an enormous amount of data in this work and the assays used are sophisticated and state-of-the-art. In general, the figures are nicely prepared and convincing. The discussion is interesting. Weaknesses are few and suggestions are listed below.

**Part II – Major Issues: Key Experiments Required for Acceptance**

Reviewer #1: (No Response)

Reviewer #2: 1. Figure 1 does a good job at illustrating important aspects of the pseudopregnancy model. However, for readers who are trained in bacterial pathogenesis and not familiar with the female reproductive cycle, it might help to:

a. Define better what is going on during the decidualization phase and how it mimics what happens during pregnancy.

b. Panel D – include some arrows that point to the different cell types mentioned in the text. Are any erythrocytes visible in the picture of the menstruation phase?

2. Also, with respect to Figure 1,

a. Please state what is being compared in panels B and C in terms of the statistical differences (most likely the comparison is to MPA-treated mice, but this is not stated clearly here as it is in the other figures).

b. Please define the abbreviation “ssoil” that is used in the figure, either in the figure legend or better, write it out fully in the figure.

c. Panel H – multiple data points are shown within each bar, but the figure legend says the data are from a representative animal. This reviewer may not understand that assay, but it is wondering if each symbol corresponds from data from each animal within a group. And why are panels (B,C) mentioned on line 515. What are they referring to?

3. Statistics: A more detailed description of the statistics used should be included in the Methods section. As the manuscript stands, most all of the all figure legends state

“Models used to compare a difference of means were fit using multiple comparisons”. What statistical test was used? What software was used?

4. The use of the term(s) “circulating” or “in circulation” is not as accurate as saying plasma level (at least occasionally), in part so the reader knows what was measured. Example, lines 1222 … “progesterone levels in circulation had reached …” would be better stated as “plasma progesterone levels had reached…”

5. The Figure legends need figure titles instead of starting with each panel. For example, for Figure 2 – the figure title should say that the data are from mouse tissue and what the experiment(s) was in general (i.e. looking atr cell distribution with respect to anatomic site and cell types and cytokine levels with respect to day of sampling). For Figure 3, a figure title should say that the data are from the examination of naturally menstruating macaques… etc.

6. Figure 4: Are these data from two separate experiments or one experiment with the same MPA control data being plotted in both panel B and panel C on different scales? If the results are from one experiment, all the data should be shown in one figure.

**Part III – Minor Issues: Editorial and Data Presentation Modifications**

Reviewer #1: (No Response)

Reviewer #2: - Why is progesterone and estrogen always capitalized within a sentence? This is distracting.

- Line 37: “murine model of menstruation to investigate how endometrial shedding and repair alter the…” - alter should be alters

- Lines 75-78 “A major challenge in studying how menstruation impacts immune defenses at barrier sites is a lack of model systems that menstruate [15], especially common laboratory animal models that are supported by immunologic and genetic approaches that could facilitate mechanistic investigations into the dynamics of FRT tissue-localized immune cell populations [16-22]”. This is a beautiful concept but the sentence is long and complicated (could be two sentences) and the “supported by” phrase isn’t clear. Perhaps “especially common laboratory animal models with which immunological an genetic approaches can be used to facilitate….”

-Line 92 C. muridarum (C. muridarum), a murine strain of chlamydia which models lower FRT infection by C. trachomatis [26, 27]”. The C. muridarum/mouse infection models upper FRT infection also, and very well compared to Ct in mice, including damage to the oviducts.

- Line 97 “…we found that while challenges administered under conditions of uterine repair resulted in robust infections, whereas challenges administered during…” – delete either “while” or “whereas”

PLOS authors have the option to publish the peer review history of their article (what does this mean? ). If published, this will include your full peer review and any attached files.

**Do you want your identity to be public for this peer review?** For information about this choice, including consent withdrawal, please see our Privacy Policy .

Reviewer #1: No

Reviewer #2: No
---

## [Decision Letter · Decision Letter 1]

Dear Dr. Kohlmeier,

We are pleased to inform you that your manuscript 'Murine modeling of menstruation identifies immune correlates of protection during Chlamydia muridarum challenge.' has been provisionally accepted for publication in PLOS Pathogens.

Best regards,

Jorn Coers

Academic Editor

PLOS Pathogens

Matthew Wolfgang

Section Editor

PLOS Pathogens

Sumita Bhaduri-McIntosh

Editor-in-Chief

PLOS Pathogens

orcid.org/0000-0003-2946-9497

Michael Malim

Editor-in-Chief

PLOS Pathogens

orcid.org/0000-0002-7699-2064

Reviewer Comments (if any, and for reference):

Reviewer's Responses to Questions

**Part I - Summary**

Reviewer #2: This manuscript has been strengthened by the conscientious response of the authors to the critiques. Strengths of the paper continue to be the use of two different animal models and appropriate experimental methods to identify protective immune cell populations in the FRT that that impact susceptibility to chlamydial infection during the menstrual cycle. Improvements to this manuscript include revision of many figures to increase the clarity of the approaches used and the information obtained, and in many cases to addition of new data in response to reviewers’ concerns. This reviewer finds the new data added to Figure 6 that show a difference in the transcriptional profiles of FRT NK cells from MPA-treated mice versus FRT NK cells from day 8 of pseudopregnancy to be convincing and also interesting in terms of its mechanistic implications.

**Part II – Major Issues: Key Experiments Required for Acceptance**

Reviewer #2: No major issues

**Part III – Minor Issues: Editorial and Data Presentation Modifications**

Reviewer #2: No minor issues

PLOS authors have the option to publish the peer review history of their article (what does this mean? ). If published, this will include your full peer review and any attached files.

**Do you want your identity to be public for this peer review?** For information about this choice, including consent withdrawal, please see our Privacy Policy .

Reviewer #2: **Yes: ** Ann Jerse

---

## [Editor Report · Acceptance letter]

Dear Dr. Kohlmeier,

We are delighted to inform you that your manuscript, "Murine modeling of menstruation identifies immune correlates of protection during Chlamydia muridarum challenge.," has been formally accepted for publication in PLOS Pathogens.

Best regards,

Sumita Bhaduri-McIntosh

Editor-in-Chief

PLOS Pathogens

orcid.org/0000-0003-2946-9497

Michael Malim

Editor-in-Chief

PLOS Pathogens

orcid.org/0000-0002-7699-2064